# Human T-cell lymphotropic virus type 1 transmission dynamics in rural villages in the Democratic Republic of the Congo with high nonhuman primate exposure

Megan Halbrook[1], Adva Gadoth[1], Anupama Shankar[2], HaoQiang Zheng[2], Ellsworth M. Campbell[2], Nicole A. Hoff[1], Jean-Jacques Muyembe[3‡], Emile Okitolonda Wemakoy[4], Anne W. Rimoin[1]*, William M. Switzer[2]*

1 University of California Los Angeles, Fielding School of Public Health, Los Angeles, California, United States of America, 2 Laboratory Branch, Division of HIV/AIDS Prevention, National Center for HIV/AIDS, Viral Hepatitis, STD, and TB Prevention, Centers for Disease Control and Prevention, Atlanta, Georgia, United States of America, 3 Institut National de Recherche Biomédicale, Kinshasa, Democratic Republic of the Congo, 4 Kinshasa School of Public Health, University of Kinshasa, Kinshasa, Democratic Republic of the Congo

‡ Unavailable
* arimoin@ucla.edu (AWR); bswitzer@cdc.gov (WMS)

**Data Availability Statement:** All data cannot be shared publicly. We use sensitive patient data

## Abstract

The Democratic Republic of the Congo (DRC) has a history of nonhuman primate (NHP) consumption and exposure to simian retroviruses yet little is known about the extent of zoonotic simian retroviral infections in DRC. We examined the prevalence of human T-lymphotropic viruses (HTLV), a retrovirus group of simian origin, in a large population of persons with frequent NHP exposures and a history of simian foamy virus infection. We screened plasma from 3,051 persons living in rural villages in central DRC using HTLV EIA and western blot (WB). PCR amplification of HTLV *tax* and LTR sequences from buffy coat DNA was used to confirm infection and to measure proviral loads (pVLs). We used phylogenetic analyses of LTR sequences to infer evolutionary histories and potential transmission clusters. Questionnaire data was analyzed in conjunction with serological and molecular data. A relatively high proportion of the study population (5.4%, n = 165) were WB seropositive: 128 HTLV-1-like, 3 HTLV-2-like, and 34 HTLV-positive but untypeable profiles. 85 persons had HTLV indeterminate WB profiles. HTLV seroreactivity was higher in females, wives, heads of households, and increased with age. HTLV-1 LTR sequences from 109 persons clustered strongly with HTLV-1 and STLV-1 subtype B from humans and simians from DRC, with most sequences more closely related to STLV-1 from *Allenopithecus nigroviridis* (Allen's swamp monkey). While 18 potential transmission clusters were identified, most were in different households, villages, and health zones. Three HTLV-1-infected persons were co-infected with simian foamy virus. The mean and median percentage of HTLV-1 pVLs were 5.72% and 1.53%, respectively, but were not associated with age, NHP exposure, village, or gender. We document high HTLV prevalence in DRC likely originating from STLV-1. We demonstrate regional spread of HTLV-1 in DRC with pVLs reported to be associated with

linked to specific rural geographic locations my molecular analysis. Hence, identities of participants could be compromised. LTR and tax sequences were deposited at GenBank with the accession numbers, MT062612 - MT062720 and MT062721 - MT062830, respectively.

**Funding:** Funding for this work was provided to AWR by the National Institute of Allergy and Infectious Disease, Division of Infectious Diseases and Microbiology (1K01AI074810-01A1) and the Faucett Catalyst Fund. The funders had no role in study design, data collection and analysis, decision to publish, or preparation of the manuscript.

**Competing interests:** The authors have declared that no competing interests exist. Author Jean-Jacques Muyembe was unable to confirm their authorship contributions. On their behalf, the corresponding author has reported their contributions to the best of their knowledge.

HTLV disease, supporting local and national public health measures to prevent spread and morbidity.

## Author summary

HTLV-1 is a human retrovirus of zoonotic simian origin that has spread globally causing inflammatory and carcinogenic disease. We previously showed that persons with high nonhuman primate (NHP) exposure in the Democratic Republic of the Congo (DRC) can be infected with another simian retrovirus, simian foamy virus (SFV), suggesting they are also at risk for infection with STLV-1. We conducted follow-up analysis of the same persons from rural villages of central DRC to determine exposure and transmission risks for STLV-1 and HTLV-1 infection. Most persons, especially women, reported high levels of NHP exposure. We identified possible introduction and spread of STLV-1 from a local monkey species across households, villages, and health zones in DRC. Most persons had HTLV-1 levels above those reported in previous studies for persons with disease. Our findings reveal that coordinated public health strategies are needed at both local and national levels to prevent further spread and morbidity of HTLV-1.

## Introduction

Human T-cell lymphotropic virus type 1 (HTLV-1), the first discovered human retrovirus, is widespread globally and can be highly oncogenic in some infected persons [1,2]. With increased human mobility and migration and limited communication regarding effective prevention strategies, HTLV-1 remains a significant public health threat [3, 4]. Furthermore, two new HTLV groups were discovered recently and their epidemiology is poorly understood [5–8]. Thus, a deeper understanding of the epidemiology of HTLV is increasingly important, with leading HTLV scientists and public health experts calling for renewed efforts to eradicate HTLV [3].

In addition to HTLV-1, three other HTLV groups (types 2–4) have been identified, all four with varying geographic prevalence, disease potential, and evolutionary history [5, 9–12]. Globally, about 10–20 million people are estimated to be infected with HTLV-1, of which 2–7% progress to adult T-cell leukemia/lymphoma (ATLL) or HTLV-1 associated myelopathy/tropical spastic paraparesis (HAM/TSP) [4, 13]. Although HTLV-1 can be found worldwide, its distribution is heavily clustered in specific populations [14, 15]. Of the seven described HTLV-1 subtypes, five (HTLV-1B, D, E, F, and G) are found primarily in Central Africa [2, 16]. HTLV-2 has also spread worldwide but is endemic in Amerindians and in persons who inject drugs. HTLV-2 has a lower pathogenicity than HTLV-1 but has been associated with various inflammatory diseases [17]. HTLV-3 and -4 were identified more recently in West-Central Africa in nonhuman primate (NHP) hunters and the pathogenicity and transmission potential of these viruses is not yet well understood [8, 18–21].

Molecular characterization of primate T-lymphotropic viruses (PTLVs), which consists of both simian and human T-lymphotropic viruses, reveals that some HTLV subtypes share closer genetic ties to certain simian T-lymphotropic viruses (STLVs) than to other HTLV subtypes, suggesting sustained zoonotic transmission of STLV between nonhuman primates (NHPs) and humans [12]. Phylogenetic and epidemiologic evidence from Central Africa supports the notion that, in addition to known human transmission pathways of HTLV (sexual, mother-to-child, sharing of needles, transplantation of infected tissues, and blood transfusions), crossover events of interspecies transmission of STLV to humans have occurred [4, 7, 12, 14]. Since HTLV testing is not routinely employed in most countries except for blood

banks in developed nations, asymptomatic carriers can unknowingly transmit HTLV both vertically and horizontally. HTLV-1, HTLV-3, and HTLV-4 have all been shown to originate from closely related STLVs (STLV-1, STLV-3, and STLV-4, respectively), whereas HTLV-2 is more distantly related to STLV-2 making its origin less clear. Phylogenetic analysis shows that all HTLV-1 subtypes except for cosmopolitan subtype A likely have primate origins [22].

Regions of frequent and close contact with wild animals have been implicated in the initiation and propagation of major infectious disease zoonoses throughout history, highlighted by the current COVID-19 pandemic [23–25]. In settings where environmental barriers between humans and animal habitats are diminished, exposure to the tissues and bodily fluids of wild animals can lead to the spread of zoonotic agents, including STLV. In the Democratic Republic of the Congo (DRC), an estimated 52 million people reside in rural, often densely forested areas and rely heavily on the hunting, trading, and consumption of bushmeat including that from NHPs, as a major source of nutrition and income [26–28]. Studies from densely forested populous areas of this Central African nation estimate that small diurnal monkeys, specifically *Cercopithecus spp.* and *Cercocebus spp.* are preferred protein sources and account for around one third of the bushmeat market [29, 30]. The encroachment of local populations on these forests of the biodiverse Congo Basin for nutritional and economic supplementation provides opportunities for cross-species transmission via bodily fluid exchange with smaller species as well as NHPs, increasing the probability of novel HTLV emergence [9, 10, 19]. Divergent STLV-1, STLV-2, and STLV-3 have been reported in monkeys and apes in DRC, combined with the hunting and eating of NHPs in this area, increases the likelihood of exposure to these viruses [31–34]. In fact, one recent study reported an association of severe NHP bites with HTLV-1 infection and showed that a significant number were genetically related to STLV-1 from gorillas and monkeys, though human-to-human transmission could not be excluded [33]. Concerningly, high-risk exposures to multiple species of wild animals has been observed in the DRC and other ecologically/anthropologically similar regions of Central Africa, making this region a hotspot for continued pathogen spillover with potential novel disease initiation events [35–38].

Despite the ubiquity of close animal contact throughout DRC and other parts of Central Africa, little evidence exists to describe the contact type most likely to result in increased zoonotic transmission of PTLV infection, based on species encountered, animal interactions, and contact frequency [32, 33, 39, 40]. Data to examine the risk of familial and intra-household transmission of HTLV once STLV crosses over into humans are similarly sparse, hampering our understanding of secondary transmission of HTLV [41, 42]. To better understand these zoonotic and person-to-person transmission pathways, we conducted a population-based survey among residents from two health zones in the rural Sankuru province of the DRC to assess the prevalence, epidemiologic risk factors, and markers of HTLV infection in this highly bushmeat-exposed population. Previously, we have shown that persons exposed to NHPs in this population were infected with simian foamy virus (SFV), another simian retrovirus, highlighting their risk for exposure to additional simian retroviruses [36]. In our current study, we aimed to characterize the transmission dynamics of HTLV, and to evaluate pathways for STLV zoonotic transmission from animals to humans and from person-to-person (sexual, vertical) in this rural Congolese population.

## Methods

### Ethics statement

The UCLA Institutional Review Board (IRB #10-000094-CR-00009) and the Kinshasa School of Public Health Ethics Committee approved collection, storage, and future testing of blood

samples collected in 2007 from all consenting study participants. A non-research determination was approved for retrovirus testing of anonymized samples at CDC.

## Study population

This study was originally conceived of as a means of conducting zoonotic surveillance of monkeypox disease in DRC, and the study design and population have been described [35, 36]. Briefly, we conducted a population-based survey in rural villages of Sankuru province, DRC from August to September 2007. Two monkeypox-endemic health zones within Sankuru province, Kole and Lomela, were chosen for study activities; village lists of these two health zones provided by local officials were used to randomly select 9 villages as study sites (Fig 1). All healthy individuals ≥ 1 year of age in selected villages were eligible for enrollment. Local,

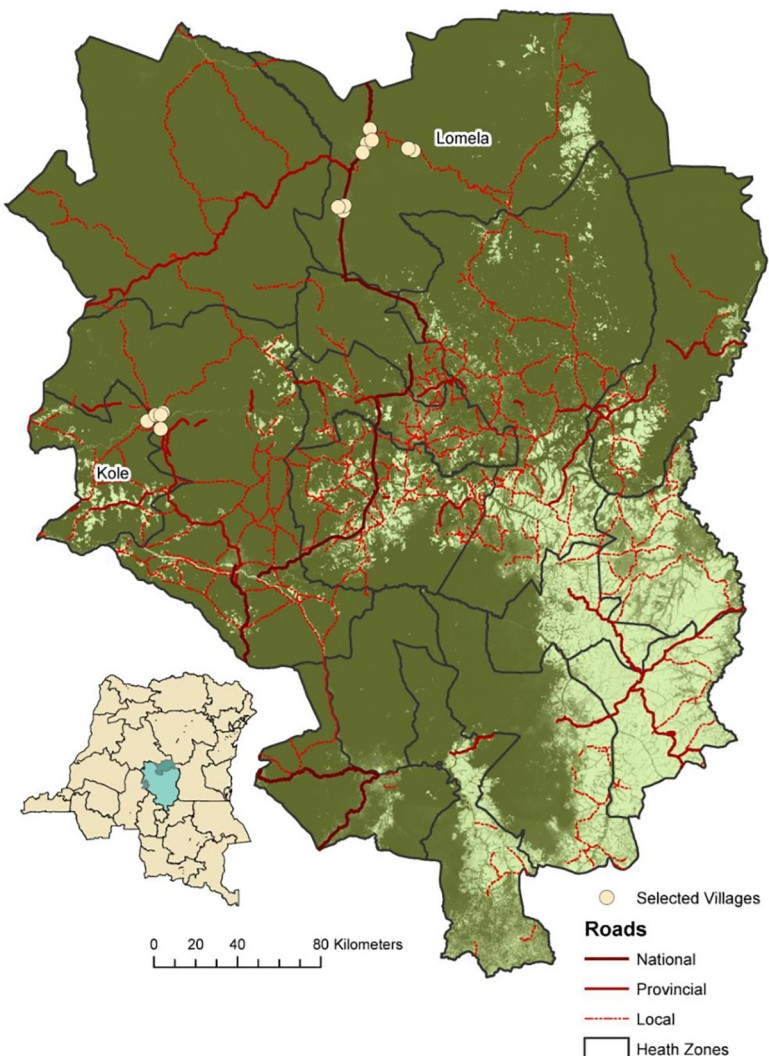

**Fig 1. Map showing the location of the 14 rural villages in the Lomela and Kole health zones in the Sankuru District of the Democratic Republic of Congo (DRC).** Small map on left is DRC with the Sankuru District in green in the center; large map on right is an expanded view of the Sankuru District. Dark green indicates forested regions and light green is savanna, yellow circles are the 14 rural villages. National, provincial, and local roads are shown in dark red, red, and red dashed lines, respectively.

trained health care workers obtained verbal informed consent from all participating adults and assent from children 7–18 years with parental or guardian consent and administered a questionnaire in either French or the local language, Tetela. Consenting parents and guardians of participants <7 years of age answered on behalf of their children. All participant data was anonymized using a unique ID number of randomly assigned check digits, which was attached to both survey data and biological samples.

## Questionnaire administration

We collected socio-demographic information via an orally administered questionnaire for each participant. Household information, including location of household and an individual's role within the household was also collected, with household role categorized according to each respondent's relationship to their respective head of household. Animal exposure data was collected with special care taken to reduce misclassification by translating scientific taxa to local nomenclature. We used focus groups to identify local names of commonly hunted animals in the region and created a handout with representative photos or drawings of each species to aid in bushmeat classification and identification. Participants were asked about the frequency and types of exposures they may have had to over 26 different animal species in the past month, including 11 NHPs found in the Sankuru province of DRC, and they also had the opportunity to specify additional species not included in the standard curated list (See S1 Table for full list of animal species). All surveys were administered by local, trained interviewers. To minimize bias that may be associated with unauthorized hunting activities, questions regarding animal exposures were never prompted or asked in a framework of legality, nor were certain species grouped by their vulnerability or conservation status.

## Biological specimen collection and laboratory analysis

Venous blood specimens were collected by trained phlebotomists from all consenting participants using ethylenediamine tetra acetic acid (EDTA)-treated vacutainer tubes (Fisher Scientific, Pittsburgh, PA). Blood specimens were processed for plasma and buffy coats in DRC, stored at -80˚C, and sent to collaborating laboratories at the US National Institutes of Health before being sent to the US Centers for Disease Control and Prevention (CDC) for final analysis.

## Serology, PCR, and phylogenetic analysis

Plasma samples were screened for the presence of antibodies to HTLV using a commercial enzyme-linked immunosorbent assay that contains antigens for both HTLV-1 and -2 (HTLV-I/II ELISA 4.0; MP Biomedicals, Santa Ana, CA). Samples with optical density values above the assay cut-off determined using the manufacturer's instructions were re-tested in duplicate, and repeat reactivity was confirmed by western blot (WB) testing that also contains antigens for both HTLV-1 and -2 (HTLV Blot 2.4; MP Biomedicals, Santa Ana, CA). The WB banding pattern of antibodies against HTLV type-specific recombinant peptides (rgp46-I, rgp46-II), Env peptide (GD21) and Gag (p24) protein were used to classify specimens as HTLV-1, HTLV-2, untypeable, indeterminate, or negative [7]. Seropositive specimens that were reactive to rgp46-I (MTA-1) or rgp46-II (K55) were considered HTLV-1-like or HTLV-2-like, respectively. Seropositive samples not reactive to either the MTA-1 or K55 peptides or to both were considered HTLV-positive, but untypeable. Specimens that were reactive to either p24 or GD21 alone or in combination with other HTLV proteins (p19, p26, p28, p32, p36, gp46, and p53) (Fig 2) were considered indeterminate. Samples with HTLV-1 Gag indeterminate patterns (HGIP) were defined as having reactivity to Gag p19, p26, p28, p32, without

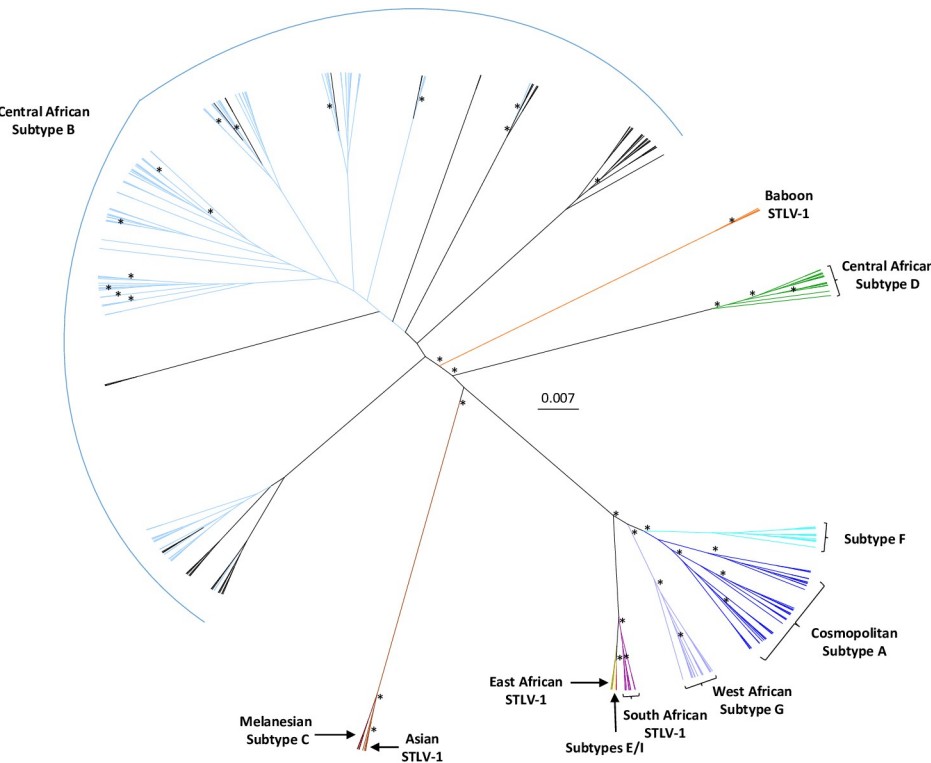

**Fig 2. Unrooted radial tree phylogeny of long terminal repeat (LTR) sequences from the Democratic Republic of Congo (DRC) using Bayesian inference.** Reference sequences were obtained by BLAST analysis of the GenBank database and selected HTLV-1 and STLV-1 subtype sequences were included in the analysis. The final alignment consisted of 273 taxa with a length of 645-nt with gaps. Posterior probabilities ≥ 0.8 are indicated by an asterisk. Trees were displayed using FigTree v1.4.0. Study sequences are colored light blue and cluster with Central African Subtype B sequences.

reactivity to Gag p24 and *env* glycoproteins (gp21, K55 and MTA-1) [43]. Persons with HTLV WB-positive results were considered infected regardless of PCR status. This combination of EIA and WB testing has been previously used to identify divergent PTLVs, including PTLV-1, -2, -3, and -4 [7, 8, 44–46].

WB-reactive samples with matching buffy coat specimens were tested for PTLV sequences using nested PCR and quantitative PCR (qPCR). DNA was extracted from archived buffy coats using the FlexiGene DNA extraction kit (QIAGEN Inc., Germantown, MD) and quantified using a NanoDrop microvolume spectrophotometer (ThermoFisher Scientific, Waltham, MA) that is calibrated at least annually. 0.5 ug of genomic DNA was used as input into a semi-nested PCR reaction using primers specific to the conserved *tax* region of PTLV to determine PTLV subgroup via sequence analysis. We and others have used generic *tax* PCR to detect highly divergent PTLV [7, 8, 20, 32, 44–48]. PH1F (5' TTG TCA TCA GCC CAC TTC CCA GG 3') and PH2R (5' AAG GAG GGG AGT CGA GGG ATA AGG 3') are the primary PCR primers and the nested PCR primers are PH2F (5' CCC AGG TTT CGG GCA AAG CCT TCT 3') and PH2R, yielding a 222-bp product [7, 49]. We also used nested PCR for some *tax* sequences using the primary primers PH1FN (5' YYI TCA GCC CAY TTY CCA GG 3') and AV46 (5' KGG RGA IAG YTG GTA KAG GTA 3') and secondary primers PH2FN (5' YCC AGG ITT YGG RCA RAG YCT YCT 3") and AV43 (5' AV43, CCA SRK GGT GTA IAI GTT TTG G 3') to generate a 430-bp amplicon [50]. Long terminal repeat (LTR) sequences (692-bp) were detected using PTLV-1-specific primers [7, 12, 51]. Spectrophotometry and

β-actin PCR were used to confirm the quality of the extracted DNA as previously reported [7, 52].

200 ng of DNA from HTLV-1-positive specimens was also tested using a generic *tax* quantitative PCR (qPCR) assay to measure proviral load (pVL). The qPCR assay used the AgPath-ID one step RT-PCR kit (Applied Biosystems, Foster City, CA) and primers PTLV-UNV-F (5' CTG GGA CCC CAT CGA TGG A 3') and PTLV-UNV-R (5' GGG GTR AGR ACY TTG AGG GT 3') and TaqMan probe PTLV-UNV-PR (5' TCK YTG GGT GGG GAA GGA GGG GAG 3'). This assay has been shown to have a linear range of seven logs and can reliably detect at least 10 copies of PTLV-1, -2, -3, and -4 per reaction [38]. Ribonuclease P protein subunit p30 (RPP30) sequences were detected in the same assay using the primers and probe RPP30 FM (5' GCA GAT TTG GAC CTG CGA GCG 3'), RPP30 RM (5' GTG AGC GGC TGT CTC CAC AAG 3') and RPP30 PM (5' FAM-TTC TGA CCT GAA GGC 'T'CT GCG CGG 3'), respectively. Percentage of HTLV-1 per 100 infected cells was calculated using the formula [PTLV *tax* copies/(0.5 x RPP30 copies)] x 100 as described previously [53].

Amplicons from the *tax* and LTR PCRs were purified using the Qiaquick PCR purification kit (QIAGEN Inc., Germantown, MD) and sequenced using the Big Dye Terminator Reaction Mix (Applied Biosystems) and an ABI 3500 sequencer. Sequences were assembled using Geneious v 9.0. Genetically related sequences were identified using BLAST searches (http://www.ncbi.nlm.nih.gov/BLAST/) and added to the analyses for comparison along with HTLV references, HTLV sequences previously isolated from DRC, and STLV sequences from various species native to the region. As with other HTLV molecular epidemiology studies, the inclusion of other DRC and African sequences in the phylogenetic analyses helps to elucidate their origin of transmission and evolutionary histories [7, 14, 32, 46, 51, 52, 54–60]. We kept the top ten PTLV sequences identified by the BLAST search and then removed any duplicates and LTR sequences < 400 nucleotides in length. For example, STLV-1 sequences (n = 34) recently reported from a variety of NHPs in DRC, including *Allenopithecus nigroviridis*, *Cercopithecus ascanius*, *C. denti*, and *C. mitis*, were excluded from our analysis since these sequences overlapped our HTLV-1 sequences and most other PTLV-1 sequences at GenBank by < 220-nt and hence were not sufficient for phylogenetic analysis [34]. In addition, the lack of phylogenetic signal in these short *tax* sequences in the alignment was confirmed using likelihood mapping analysis in IQ-Tree v1.6.0 [61]. We also limited the number of HTLV-1 sequences from a specific study and country, except DRC, to 2–4 taxa to reduce the computational complexity of the Bayesian analysis. We performed DNA alignments using MAFFT v7.017. GUIDANCE2 was used to identify and remove phylogenetically unreliable regions in the alignment at the recommended confidence score of 0.93 [62]. All LTR sequences in the final alignment passed both the composition Chi square test and likelihood mapping analysis in IQ-Tree. We inferred HTLV-1 LTR phylogenies using Bayesian inference using BEAST v1.8.4 [63] with an uncorrelated, lognormal relaxed molecular clock, a birth-death tree prior and 450 million Markov Chain Monte Carlo (MCMC) iterations with a 10% burn-in. These parameters have been shown previously to accurately infer PTLV evolutionary histories [56, 60, 64]. Convergence of the MCMC was assessed by calculating the effective sampling size (ESS) of duplicate runs using the program Tracer v1.6 (http://tree.bio.ed.ac.uk/software/tracer/). We used the model test algorithm in MEGA v6 to determine the best fitting nucleotide substitution model, which was inferred to be the general time reversible (GTR) model with gamma (G) distribution (GTR+G). An xml file is provided in the supplementary material (S1 BEAST) which includes the sequences and parameters for the BEAST analysis. All parameter estimates showed ESSs > 750 indicating sufficient mixing. The tree with the maximum product of the posterior clade probabilities (maximum clade credibility (MCC) tree) was chosen from the posterior distribution of 9,001 sampled trees after burning in the first 1,000 sampled trees with the program

Tree Annotator. Branch support was determined using posterior probabilities. Trees were displayed and edited in FigTree v 1.4.0 (http://tree.bio.ed.ac.uk/software/figtree/). LTR sequences were considered as belonging to transmission clusters if the inferred tree node posterior probabilities were > 0.8 and the members of the cluster shared > 99% nucleotide identity.

## Statistical analyses

We used t-test, Wald chi-square analyses, and Fisher's exact test to assess differences in HTLV WB positivity, demographic, behavioral, NHP exposure groups, and differences in pVLs. Wald chi-square tests of proportions were also used to determine differences in demographic breakdown between those with and without eligible biological specimens. From the serologic assays, HTLV WB positivity was classified for EIA-reactive samples showing HTLV-1-like, HTLV-2-like, or untypeable profiles. Samples with indeterminate WB results were classified as negative since these results are rarely from persons with HTLV infection [43, 49, 65]. Stratified and adjusted analyses were performed via logistic regression to compare the magnitude and significance of various demographic and behavioral risk factors on HTLV seroreactivity. Only households with at least one HTLV seroreactive individual and complete, identifiable household information were included in household-centered analyses (n = 534 individuals in 217 households). All statistical computations were performed in SAS version 9.3 (SAS Institute, Cary, NC).

## Results

### Participant sociodemographics and HTLV serology

In total, 4,572 of 5,687 (80.4%) eligible persons were enrolled in the study from 9 villages in Lomela and five villages in Kole (Fig 1). Of these enrollees, 3,071 (67%) had biological specimens of sufficient quality for testing. We found 281/3,071 (9.15%) samples were repeat reactive by ELISA, of which 172 were seropositive by WB, indicating an HTLV seroprevalence of 5.6% in this population. Of these, 135 (78.56%) were HTLV-1-like, three (1.8%) were HTLV-2-like, and 34 (19.8%) were HTLV-positive but untypeable. HTLV-indeterminate profiles were seen in 80 (2.6%) samples (Table 1). The majority of samples with untypeable WB results (58.8%,

**Table 1. HTLV Western blot (WB) and PCR results.**

| WB results | | PCR results[1] | | |
|---|---|---|---|---|
| WB profiles | Number (%) | *tax* (%) | LTR (%) | Total (%) |
| HTLV-1-like | 135/281 (48.0) | 90/125 (72.0) | 89/125 (71.2) | 90/125 (72.0) |
| HTLV-2-like | 3/281 (1.1) | 1/3 (33.3)[2] | 1/3 (33.3)[2] | 1/3 (33.3)[2] |
| HTLV untypeable | 34/281 (12.1) | 12/34 (35.3) | 12/34 (35.3) | 12/34 (35.3) |
| Indeterminate | 74/281 (26.3) | 4/69 (5.8) | 4/69 (5.8) | 4/69 (5.8) |
| HGIP[3] | 6/281 (2.1) | 0/5 (0) | 0/5 (0) | 0/5 (0) |
| EIA reactive, WB not done[4] | 7/281 (2.5) | 3/6 (42.9) | 3/6 (42.9) | 3/6 (42.9) |
| Negative | 22/281 (7.8) | ND[5] | ND | ND |
| Total | 281/281 (100) | 110/242 (45.5) | 109/242 (45.0) | 110/242 (45.5) |

1 Total differences between WB ad PCR results due to unavailability of peripheral blood mononuclear cells (PBMCs) for the PCR testing.

2 One HTLV-2 WB-positive sample (MPX29290) contained HTLV-1 sequences.

3 HGIP, HTLV Gag indeterminate WB patterns (reactivity to Gag p19, p26, p28, p32, without reactivity to Gag p24 and envelope glycoproteins (gp21, K55 and MTA-1).

4 Seven samples were HTLV EIA-reactive but insufficient plasma volumes were available for WB testing. Six of the seven had PBMCs archived for PCR testing.

5 ND, PCR testing not done.

20/34) had dual reactivity to both K55 and MTA-1 suggestive of dual HTLV-1/HTLV-2 infection as well as to the other Gag proteins (p19, p26, p28, p32, p36). Fifteen untypeable samples (42.9%) did not have reactivity to either gp46 protein or the other Gag proteins except to p19 for four samples. Six (7.1%) of the samples with indeterminate results were HGIPs. Two of the three samples with HTLV-2-like WB profiles showed reactivity to GD21, p24, p28, p36, p53 and K55, whereas the third had reactivity to only GD21, p24, and K55. Eleven persons with archived plasma specimens did not have survey data and thus our epidemiologic sample comprised of 3,060 persons (66.9%) for whom both questionnaire and serologic data were available for statistical analysis (Table 2). HTLV infection was strongly associated with both sex and age at statistically significant levels. The odds of HTLV seropositivity were 2.13 times greater for females than for males (95% CI: 1494–3.03) and on average seroreactive individuals were 14 years older than seronegative individuals (mean age of 37.9 and 23.4, respectively; $p < .0001$). The odds of HTLV seropositivity increased by 1.2 (95% CI: 1.15–1.24) for every five years of age. The average age among women was 25.4 years (range: 1–99 years) while the average age for men was 22.6 (range: 0–98 years).

Three individuals had HTLV-1-like WBs and were also found to be positive for simian foamy virus (SFV) antibodies and sequences in our previous study [36], indicating dual retroviral infection. Fourteen other study participants were SFV seropositive but HTLV-negative. An association between SFV and HTLV co-infection was not observed (Fisher's $p$-value = 0.08995).

## Households relationships, role and HTLV WB seropositivity

To analyze potential intra-household transmission of HTLV, a sub-cohort was established of all 165 HTLV seropositive individuals and their 158 known family members. 132 households had just one HTLV seropositive individual, 12 households had two, and four households had three seropositive individuals residing within them. Across the 11 different household roles for which we collected data, 84% of HTLV seropositive persons were either the head of household (HH), a primary or secondary wife (PW or SW, respectively), or a biologically related child (e.g. the nuclear family).

## NHP exposure

NHP exposures were divided into two analytical frameworks: exposures based on animal species and exposures based on activity type. For the former, any type of exposure activity with each of nine distinct NHP species was assessed: *Cercocebus chrysogaster*, *Cercopithecus ascanius*, *Cercopithecus neglectus*, *Cercopithecus nictitans*, *Cercopithecus wolfii*, *Colobus angolensis*, *Lophocebus aterrimus*, *Piliocolobus tholloni*, and *Pan paniscus*. For the latter framework, exposure to any NHP during each of six distinct exposure activities was examined: hunting, picking up dead animals, butchering and skinning, cooking, eating, and playing with or being bitten or scratched by a live animal. Chi-squared analysis showed no association between HTLV seropositivity and exposure to any individual NHP species nor to any NHP exposure type. Rates of any NHP contact were similar at each age category ($p$ = 0.1652) and were between 66.7–83.9%. While the majority of NHP activities had odds of HTLV seropositivity with low $p$ values, the odds ratios crossed the null hypothesis (1.0) and were not considered significant (Table 3). NHP exposure activities were also heavily gendered. Over 95% of hunters were men, whereas 76.7% of those reporting cooking NHPs were women. Women were more likely to be exposed to NHPs in general, and were also found to cook or eat NHPs significantly more than men (χ2 $p$ value = 0.0006, < .0001, 0.0051, respectively), whereas men were more likely to hunt or pick up dead NHPs (χ2 $p$ -value < .0001 for both) (Table 3). The frequencies of men

**Table 2. HTLV frequency among study participants from Kole and Lomela Health Zones, Sankuru Province, DRC.**

| Variable | HTLV Serostatus | | All Study Enrollees | |
|---|---|---|---|---|
| | Positive[1] n = 172 (%) | Negative n = 2899 (%) | With eligible specimens n = 3060 (%) | Without eligible specimens n = 1512 (%) |
| **HTLV** | | | | |
| HTLV-1 | 135 (78.5) | 0 (0) | | |
| HTLV-2 | 3 (1.7) | 0 (0) | | |
| Untypeable | 34 (19.8) | 0 (0) | | |
| Indeterminate | 0 (0) | 80 (2.8) | | |
| Negative | 0 (0) | 2819 (97.2) | | |
| **Healthzone[2]** | | | | |
| Lomela | 61 (35.5) | 958 (33) | 1019 (33.2) | 354 (23.4) |
| Kole | 100 (58.1) | 1905 (65.7) | 2005 (65.3) | 580 (38.4) |
| *missing* | 11 (6.4) | 36 (1.2) | 47 (1.5) | 578 (38.2) |
| **Age[2,3]** | | | | |
| *mean (range)* | $\mu = 37.9$ (2–89) | $\mu = 23.5$ (0–99) | $\mu = 24.2$ (0–99) | $\mu = 19.04$ (1–86) |
| 0–5 | 4 (2.3) | 333 (11.5) | 337 (11) | 455 (30.1) |
| 6–17 | 27 (15.7) | 1037 (35.8) | 1064 (34.6) | 417 (27.6) |
| 18–25 | 26 (15.1) | 466 (16.1) | 492 (16) | 175 (11.6) |
| 26–40 | 29 (16.9) | 517 (17.8) | 546 (17.8) | 217 (14.4) |
| 41–59 | 49 (28.5) | 394 (13.6) | 443 (14.4) | 151 (10) |
| 60–99 | 30 (17.4) | 136 (4.7) | 166 (5.4) | 62 (4.1) |
| *missing* | 7 (4.1) | 16 (0.6) | 23 (0.7) | 35 (2.3) |
| **Sex** | | | | |
| Female | 122 (70.9) | 1649 (56.9) | 1771 (57.7) | 830 (54.9) |
| Male | 43 (25) | 1236 (42.6) | 1279 (41.6) | 649 (42.9) |
| *missing* | 7 (4.1) | 14 (0.5) | 21 (0.7) | 33 (2.2) |
| **Household Role** | | | | |
| Head of Household (HH) | 47 (27.3) | 572 (19.7) | 619 (20.2) | 190 (12.6) |
| Primary Wife (PW) | 24 (14) | 262 (9) | 286 (9.3) | 89 (5.9) |
| Secondary Wife (SW) | 27 (15.7) | 285 (9.8) | 312 (10.2) | 86 (5.7) |
| Biological Child (BC) | 24 (14) | 905 (31.2) | 929 (30.3) | 239 (15.8) |
| Other Child | 1 (0.6) | 22 (0.8) | 23 (0.7) | 7 (0.5) |
| Grandchild (GC) | 2 (1.2) | 50 (1.7) | 23 (0.7) | 6 (0.4) |
| Sibling (Sib) | 10 (5.8) | 64 (2.2) | 74 (2.4) | 12 (0.8) |
| Sibling-in-law | 0 (0) | 33 (1.1) | 33 (1.1) | 13 (0.9) |
| Uncle or Aunt | 4 (2.3) | 8 (0.3) | 12 (0.4) | 5 (0.3) |
| Parent (P) | 4 (2.3) | 78 (2.7) | 82 (2.7) | 22 (1.5) |
| Grandparent | 3 (1.7) | 20 (0.7) | 52 (1.7) | 8 (0.5) |
| Other Family Members | 0 (0) | 3 (0.1) | 3 (0.1) | 0 (0) |
| Not Family | 0 (0) | 2 (0.1) | 2 (0.1) | 0 (0) |
| *missing* | 26 (15.1) | 595 (20.5) | 621 (20.2) | 835 (55.2) |
| **Ethnicity[2]** | | | | |
| Batetela | 44 (25.6) | 682 (23.5) | 726 (23.6) | 213 (14.1) |
| Ohindo | 80 (46.5) | 1301 (44.9) | 1381 (45) | 346 (22.9) |
| Other[4] | 21 (12.2) | 312 (10.8) | 333 (10.8) | 115 (7.6) |
| *missing* | 27 (15.7) | 604 (20.8) | 631 (20.5) | 838 (55.4) |

1 WB, Western blot. Positivity defined as HTLV WB profiles of HTLV-1, HTLV-2, or HTLV untypeable.

2 Wald chi-square test of proportions significant at $\alpha = 0.05$ by specimen status (eligible vs ineligible)

3 Wald chi-square test of proportions significant at $\alpha = 0.05$ by HTLV status (seroreactive vs non-reactive)

4 Other ethnicities include, Bakela (n = 20), Bankutshu (n = 68), Basho (n = 12), Bambole (n = 1), and Dionga (n = 32).

and women reported butchering and skinning NHPs were nearly equal and was not a significantly different (Table 3).

## HTLV PCR and phylogenetic analyses

Two hundred and forty HTLV WB seropositive and seroindeterminate persons (96.0%) had buffy coats available for testing by nested PCR using generic *tax* primers for determination of HTLV group and by generic *tax* qPCR for simultaneous viral detection and pVL determination. All DNA specimens from the 240 buffy coats had amplifiable β-actin sequences demonstrating the integrity of the extracted nucleic acids. About 40% of the WB seroreactive specimens tested positive for *tax* sequences by either nested (n = 110) and/or qPCR (n = 107). Of the 110 *tax* PCR-positive specimens, the majority (90/110 = 81.8%) had HTLV-1 WB profiles, compared to one with an HTLV-2 WB profile (0.9%), 12 HTLV-positive but untypeable profiles (10.9%), four WB indeterminates (3.6%), and three samples with insufficient plasma volume for WB testing (2.7%). Of the 12 PCR-positive samples with HTLV-untypeable WB results, 8 had seroreactivity to both gp46 proteins and four did not. Of the 130 PCR-negative samples, 35 had HTLV-1 WB profiles (26.9%), two were HTLV-2 (1.5%), 20 were HTLV-positive but untypeable (15.4%), 70 were indeterminate (53.9), and three samples had insufficient plasma volume for WB testing (2.3%). BLAST analysis of the *tax* sequences showed that all had the highest identity to HTLV-1, of which 80 showed the highest genetic relatedness (97–100%) to two HTLV-1s from DRC (MOMS, GenBank# Y15960 and ITIS, GenBank# Y15958) [66]. One sample with an HTLV-2 WB profile had HTLV-1 *tax* and LTR sequences.

We obtained partial LTR sequences from 109 persons and used these for phylogenetic analyses. A BLAST search identified a total of 36 unique LTR sequences (28 HTLV-1, 8 STLV-1) that were genetically related to those from our study and were included with other African and subtype reference sequences (65 HTLV-1 and 63 STLV-1) for a total of 273 PTLV-1 sequences in the phylogenetic analyses. The Bayesian topology showed the LTR sequences clustered by subtype as expected demonstrating the robustness of our analysis (Fig 2), with those sequences from our study clustering in the Central African subtype B clade. Within the subtype B clade our DRC LTR sequences clustered within two separate clades with strong support (posterior probability (PP) = 1) and both contained STLV-1 sequences (Fig 3). The first clade consisted of 32 HTLV-1 and five STLV-1, including three STLV-1 from different *Cercopithecus* monkeys

**Table 3. Nonhuman primate exposure activity, gender, and association with HTLV Western blot (WB) positivity[1].**

| Activity | Total | Women | | Men | | p-value[2] | HTLV WB positive (n) | Association with HTLV | |
| --- | --- | --- | --- | --- | --- | --- | --- | --- | --- |
| | | n | % | n | % | | | Crude OR (95% CI) | Adjusted OR (95%CI)[3] |
| Any Exposure | 2007 | 1226 | 83.7 | 781 | 78.3 | 0.0007 | 111 | 0.72 (0.49–1.08) | 0.65 (0.43–0.975) |
| Hunt | 182 | 9 | 0.62 | 173 | 17.4 | < .0001 | 8 | 0.72 (0.35–1.49) | 0.66 (0.3–1.48) |
| Pick up dead | 89 | 32 | 2.19 | 57 | 5.72 | < .0001 | 5 | 0.95 (0.68–2.37) | 0.78 (0.31–2.01) |
| Cook | 1377 | 1056 | 72.1 | 321 | 32.2 | < .0001 | 91 | 1.35 (0.96–1.91) | 0.81 (0.54–1.2) |
| Butcher & Skin | 970 | 574 | 39.2 | 396 | 39.7 | 0.7988 | 57 | 1.00 (0.71–1.41) | 0.83 (0.58–1.18) |
| Ate | 1955 | 1190 | 81.3 | 765 | 76.7 | 0.0061 | 106 | 0.69 (0.47–1.00) | 0.63 (0.42–0.93) |
| Scratch | 6 | 3 | 0.02 | 3 | 0.03 | 0.4663 | 0 | - | - |
| Bite | 4 | 2 | 0.14 | 2 | 0..2 | 0.5337 | 0 | - | - |
| Play | 28 | 12 | 0.820 | 16 | 1.6 | 0.0714 | 3 | 1.94 (0.58–6.5) | 2.95 (0.85–10.2) |

1 WB positivity defined as HTLV WB profiles of HTLV-1, HTLV-2, or HTLV untypeable.

2 Chi-square or Fisher's exact test was used to calculate the *p*-value.

3 Adjusted for age and sex.

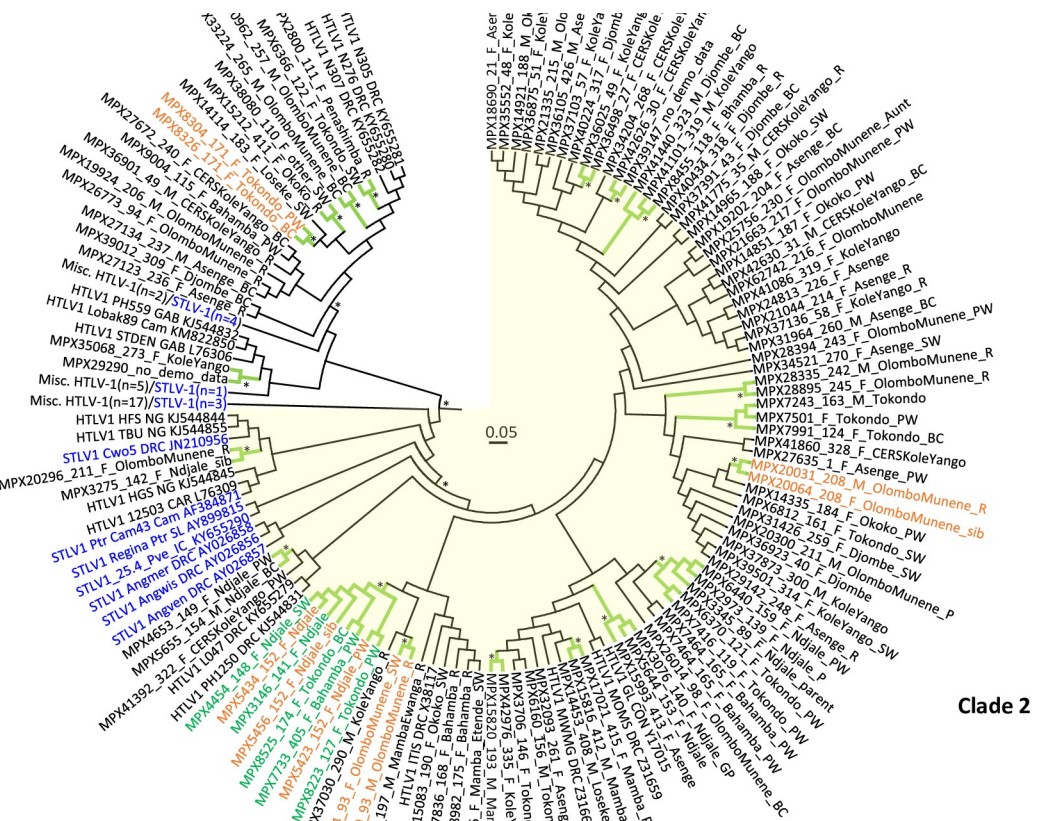

**Fig 3. Circular Bayesian phylogenetic subtree from Fig 2 containing DRC LTR sequences.** The final alignment consisted of 273 taxa with a length of 645-nt with gaps. Posterior probabilities ≥ 0.8 are indicated by an asterisk. Trees were displayed using FigTree v1.4.0. Clade 1 contains 37 taxa and is shown with a white background while clade 2 contains 127 taxa, including 85 HTLV-1 LTR sequences from our study, and is highlighted with a yellow background. The branches for those clusters with high support are colored green and marked with an asterisk. Taxa for clusters of persons belonging to the same household are colored orange and are annotated with demographics in the format: StudyID_HouseholdID_sex_village_relationship to head-of-household (HH). F, female; M, male, PW, primary wife; sib, sibling; R, responsible (same as HH); SW, secondary wife; BC, biological child; P, parent; GC, grandchild; GP, grandparent. Taxa for the eight-person cluster with the 11-bp deletion are in green; however, ones from the same household are in orange. STLV-1 taxa are blue and simian species origin are provided as three letter codes (ANG, *Allenopithecus nigroviridis*, Ggo, *Gorilla gorilla*; Ptr, *Pan troglodytes*; Pve, *P. vellerosus*; Cne, *Cercopithecus neglectus*; Cwo, *C. mona wolfii*; Cas, *C. ascanius*. Country of origin is provided in taxon name when known; Zaire is now DRC; CAM, Cameroon; IC, Ivory Coast; EG, Equatorial Guinea; CAR, Central African Republic; GAB, Gabon; NG, Nigeria; GAM, Gambia. Some branches were collapsed to improve visualization of the DRC genetic relationships. Accession numbers are provided for taxa sequences obtained at GenBank for the analysis.

from DRC (Cne8, Cwo39, Cas88) and two apes from Cameroon (PtrCAR.875, GgoGolda). Seventeen DRC LTR sequences from our study clustered with three DRC HTLV-1 sequences (N276, N305, N307) from a previous study where all three reported monkey exposure (PP = 0.89) [32]. Two additional DRC sequences from our study clustered with two HTLV-1 from Gabon (StDen, PH559) and one from Cameroon (Lobak89) [33, 51, 67]. Person Lobak89 reported a severe gorilla bite; NHP exposure information was not available for StDen and PH559.

The second clade consisted of 117 HTLV-1 and 10 STLV-1 sequences, including three STLV-1 from *Allenopithecus nigroviridis* (Angwis, Angmer, Angven) and one from a *Cercopithecus* monkey (Cwo5), both from DRC (Fig 3) [46, 58]. The remaining 7 STLV-1 were from apes from Cameroon (Cam43, GGoCam12, Ggo02Cam3157, GgoM10431) and the Ivory Coast (25.4_Pve, Regina_Ptr). Eighty-five DRC HTLV-1 from our study clustered strongly (PP = 0.97) in a subclade in the 127-member clade with four HTLV-1 from DRC (MWMG,

GL, MOMS, ITIS) from the 1990s [66, 68]. This 89-member DRC taxa clade was sister to a five-member clade containing three DRC sequences from our study (MPX4653, MPX5655, MPX41392) and two DRC HTLV-1 from a previous study (PH1250, L047) for whom NHP exposure was not reported [32, 67]. However, separation of these two DRC clades was weakly supported (PP = 0.13). Interestingly, the three *A. nigroviridis* STLV-1 were ancestral to this 89-member DRC taxa clade with good support (PP = 0.77). Two additional DRC sequences from our study (MPX3275, MPX4653) clustered in an 8-member clade with two HTLV-1 from Nigeria (TBU, HFS), one HTLV-1 from the Central African Republic (12503), an STLV-1 from a *Cercopithecus* monkey from DRC (Cwo5), and an ape from Cameroon (Ptr_Cam43).

In the BEAST analysis, we found 18 potential transmission clusters that included a total of 50 persons (Table 4). Three pairs of HTLV-1 LTR sequences, 2 persons in a three-member cluster, and three persons in an 8-member cluster all clustered with good support (PP > 0.83) and the participants in each group were from the same household (Fig 3, Table 4). Among these eleven persons from five households, just one cluster had an additional HTLV seroreactive member who was not a part of the cluster. In this house, from the village of Tokondo, a primary wife and a female biological child (MPX8304 and MPX8326) clustered with strong support and one of the three additional male biological children in this household was seroindeterminate, but PCR-negative. Four of these households had additional family members, primarily other biological children who were not positive for HTLV. In the fifth house, only the two found positive participated in the study (MPX20031 and MPX20064). Interestingly, in one cluster a primary wife and a female biological child (MPX7501 and MPX7991) clustered with a male biological child from a different household (MPX7243) despite there being two other male biological children in their household. In one 10-person household in the village of Ndjale, three women were a part of a large eight-person cluster.

It is important to note that we also found several instances (11 pairs, two triads, two five-member clusters, one 8-member cluster) where sequences from different households, and even different villages, clustered together with good phylogenetic support (Fig 3, Table 4). The DRC map (Fig 1) shows the connectivity of the villages by roadways, which for the two health zones are 130–160 km apart. The majority (8/15, 53.3%) contained only females but there were five male/female pairs, one male/male pair, and one triad consisting of two males (7 and 28 yo) and one female (78 yo). One male/female pair Cluster 13) and the two females and one male in a triad (Cluster 9) were from the same village but from different households. Both females in the triad were from the same household; one was the primary wife and her biological child. Persons in these two pairs differed in age by only four years. In one triad (Cluster 16) and in one pair (Cluster 5) at least one member was even from a village from a different health zone. Both five-member clusters 7 and 18 consisted of four females and one male and were from different villages in the Lomela health zone. The latter five-member cluster also clustered with the HTLV-1-ITIS LTR sequence from an adult male from DRC with HTLV-1-associated myelopathy/ tropical spastic paraparesis (HAM/TSP) [66].

The eight-member cluster (Cluster 8) consisted of all women of different ages from two different villages in the Lomela health zone. Examination of the HTLV-1 LTR alignment showed all eight LTRs had the same 11-bp deletion. The deletion is located just before the first transcription enhancer element in the LTR. Five women were from Ndjale, of who three (MPX5423, 5434, and 5456) were from the same household as described earlier. In relation to the head of household, MPX5423 is the primary wife and MPX5456 is a sibling; the relationship of MPX5434 is not known. Two of the eight were from Tokondo, and one was from Bahamba. The MPX8223 HTLV-1 sequence in this cluster is the only one of the three confirmed SFV-infected persons in our study whose SFV originated from an Angolan colobus monkey (*Colobus angolensis*) endemic to DRC [36].

**Table 4. Characteristics of persons clustering in potential HTLV-1 transmission networks[1].**

| Cluster ID | PersonID[2] | AGE | Sex | Ethnicity | HID[3] | Village[4] | Health Zone | Household Relationship[5] | Forest frequency | Any NHP exposure | BEAST phylo-genetic support (PP) | % PTLV tax cp/100 PBMC[6] | Cluster size |
|---|---|---|---|---|---|---|---|---|---|---|---|---|---|
| 1 | 15212 | 68 | F | Batetela | | Okoko | Lomela | HH | > 4 times/month | No | 1 | 0.1 | Pair |
| | 38080 | 45 | F | Bankutshu | 110 | Other village | Kole | Secondary wife | > 4 times/month | Yes | 1 | 0.17 | |
| 2 | 15816 | 23 | M | Other | | Mamba | | Grandchild | Daily | No | 0.97 | 1.54 | Pair |
| | 17021 | 27 | F | Other | | Mamba | | Primary wife | 2–4 times/month | Yes | 0.97 | 31.83 | |
| 3 | 15820 | 82 | M | Dionga | 193 | Mamba | Lomela | HH | Never | Yes | 0.83 | 1.27 | Pair |
| | 42976 | 43 | F | Bakela | 335 | CERS Kole Yango | Kole | HH | Daily | Yes | 0.83 | 0.35 | |
| 4 | 20031 | 32 | M | Ohindo | 208 | Olombo Munene | Kole | HH | Once/month | Yes | 0.99 | 3 | Pair |
| | 20064 | 38 | F | Ohindo | 208 | Olombo Munene | Kole | Sibling | Daily | Yes | 0.99 | 3.71 | |
| 5 | 3275 | 15 | F | Batetela | 142 | Ndjale | Lomela | Sibling | > 4 times/month | Yes | 1 | 6.99 | Pair |
| | 20296 | 47 | F | Ohindo | 211 | Olombo Munene | Kole | HH | Once/month | Yes | 1 | 71.2 | |
| 6 | 23030 | 33 | M | Ohindo | 93 | Olombo Munene | Kole | HH | > 4 times/month | Yes | 1 | 2.63 | Pair |
| | 23074 | 30 | F | Ohindo | 93 | Olombo Munene | Kole | Secondary wife | > 4 times/month | Yes | 1 | 0.66 | |
| 7 | 2973 | 49 | M | Other | 139 | Ndjale | Lomela | Responsible | 2–4 times/month | Yes | 0.97 | 0.08 | 5-member |
| | 6440 | 49 | F | Batetela | 159 | Ndjale | Lomela | Primary wife | > 4 times/month | No | 0.97 | 1.53 | |
| | 3345 | 72 | F | Batetela | | Ndjale | Lomela | | Once/month | Yes | 0.97 | 0.2 | |
| | 6370 | 55 | F | Batetela | 121 | Bahamba | Lomela | Primary wife | > 4 times/month | Yes | 0.97 | 1.41 | |
| | 7416 | 22 | F | Other | 119 | Bahamba | Lomela | Primary wife | Daily | Yes | 0.97 | 3.28 | |
| 8 | 3146 | 15 | F | | 141 | Ndjale | Lomela | | | No | 1 | 73.06 | 8-member |
| | 4454 | 25 | F | Batetela | 148 | Ndjale | Lomela | Secondary wife | > 4 times/month | Yes | 1 | 1.05 | |
| | 5423 | 32 | F | Batetela | 152 | Ndjale | Lomela | Primary wife | Daily | Yes | 1 | 6.21 | |
| | 5434 | 29 | F | | 152 | Ndjale | Lomela | | | No | 1 | 1.1 | |
| | 5456 | 15 | F | Batetela | 152 | Ndjale | Lomela | Sibling | Daily | No | 1 | 8.29 | |
| | 7733 | 33 | F | Other | | Bahamba | Lomela | Primary wife | Daily | No | 1 | 0.12 | |
| | 8223 | 23 | F | Batetela | 127 | Bahamba | Lomela | Primary wife | Daily | Yes | 1 | 1.06 | |
| | 8525 | 12 | F | Other | 174 | Bahamba | Lomela | Biological child | > 4 times/month | Yes | 1 | 4.41 | |
| 9 | 7243 | 6 | M | | 163 | Bahamba | Lomela | | | No | 1 | 0.61 | 3-member |
| | 7501 | 50 | F | Other | 124 | Tokondo | Lomela | Primary wife | Daily | Yes | 1 | 4.44 | |
| | 7991 | 11 | F | Batetela | 124 | Tokondo | Lomela | Biological child | Never | Yes | 1 | 3.98 | |
| 10 | 8304 | 49 | F | Bahamba | 171 | Tokondo | Lomela | Primary wife | Daily | No | 1 | 2.31 | Pair |
| | 8326 | 9 | F | Other | 171 | Tokondo | Lomela | Biological child | 2–4 times/month | No | 1 | 2.72 | |

(*Continued*)

**Table 4.** (Continued)

| Cluster ID | PersonID[2] | AGE | Sex | Ethnicity | HID[3] | Village[4] | Health Zone | Household Relationship[5] | Forest frequency | Any NHP exposure | BEAST phylo-genetic support (PP) | % PTLV tax cp/100 PBMC[6] | Cluster size |
|---|---|---|---|---|---|---|---|---|---|---|---|---|---|
| 11 | 6366 | 47 | F | Batetela | 122 | Tokondo | Lomela | Secondary wife | > 4 times/month | Yes | 0.93 | 3.55 | Pair |
|  | 2800 | 54 | F | Batetela | 111 | Penashimba | Lomela | HH | > 4 times/month | Yes | 0.93 | 2.61 |  |
| 12 | 33224 | 14 | M | Ohindo | 265 | Olombo Munene | Kole | Biological child | 2–4 times/month | Yes | 0.84 | 1.46 | Pair |
|  | 30962 | 37 | M | Batetela | 257 | Olombo Munene | Kole | Biological child | > 4 times/month | No | 0.84 | 0.01 |  |
| 13 | 28335 | 49 | M | Ohindo | 242 | Olombo Munene | Kole | HH | > 4 times/month | No | 1 | 2.7 | Pair |
|  | 28895 | 45 | F | Ohindo | 245 | Olombo Munene | Kole | HH | Daily | Yes | 1 | 0.01 |  |
| 14 | 36025 | 23 | F | Ohindo | 49 | Kole Yango | Kole | Sibling | 2–4 times/month | Yes | 0.92 | 0.61 | Pair |
|  | 36498 | 60 | F | Ohindo | 27 | CERS Kole Yango | Kole | Grand parent | > 4 times/month | Yes | 0.92 | 0.6 |  |
| 15 | 42626 | 55 | F | Ohindo | 30 | CERS Kole Yango | Kole | HH | 2–4 times/month | Yes | 0.89 | 3.5 | Pair |
|  | 39147 | 47 | F | Ohindo |  | Kole Yango | Kole |  | 2–4 times/month | Yes | 0.89 | 6.6 |  |
| 16 | 41440 | 7 | M | Ohindo | 323 | Djombe | Kole |  | Once/month | No | 0.8 | 0.01 | 3-member |
|  | 8455 | 78 | F | Batetela | 118 | Tokondo | Lomela | HH | 2–4 times/month | Yes | 0.8 | 0.05 |  |
|  | 41101 | 28 | M |  | 319 | Kole Yango | Kole |  |  | No | 0.8 | 2.08 |  |
| 17 | 29290 |  |  |  |  |  |  |  |  |  | 1 | 0.14 | Pair |
|  | 35068 | 2 | F |  | 273 | Kole Yango | Kole |  |  | No | 1 | 0.01 |  |
| 18 | 7836 | 28 | F | Batetela | 168 | Tokondo | Lomela | HH | Daily | Yes | 0.8 | 0.49 | 5-member[7] |
|  | 8982 | 73 | F | Other | 175 | Bahamba | Lomela | HH | 2–4 times/month | Yes | 0.8 | 2.08 |  |
|  | 15083 | 45 | F | Batetela | 190 | Okoko | Lomela | Secondary wife | > 4 times/month | Yes | 0.8 | 0.11 |  |
|  | 17360 | 51 | F | Other | 196 | Mamba Etende | Lomela | Secondary wife | > 4 times/month | No | 0.8 | 0.01 |  |
|  | 17393 | 66 | M | Batetela | 197 | Mamba Ewanga | Lomela | HH | 2–4 times/month | No | 0.8 | 0.01 |  |

1 Potential transmission clusters were identified with Bayesian phylogenetic analysis. Empty cells indicate data was not reported.

2 IDs, identification number. ID in cell with blue background is person also infected with simian foamy virus.

3 HID, household ID. Those cells with gold backgrounds are persons with linked transmission within the same household.

4 Village Mamba, includes Mamba Etende, Mamba Ewanga, and Mamba Etinda.

5 HH, head of household.

6 Percent of HTLV-1 infected peripheral blood mononuclear cells (PBMCs) based on quantification of *tax* and RPP30 sequences as described in methods. BLD, below the limit of detection of the assay.

7 This 5-member cluster also clustered with HTLV-1_ITIS from DRC.

A total of 61 persons had LTR sequences that did not cluster phylogenetically with strong support. These singletons were composed mostly of women (44/59, 74.6%), most reported NHP exposure (34/54, 62.9%), and frequented the forests often (> 4 times/month; 31/47,

65.9%). Denominators are different for each variable depending on a participant's lack of response.

## HTLV-1 proviral loads

The percentage of infected cells in 107 persons with detectable pVLs ranged from 0.01–73.06% with a mean and median of 4.38% and 1.58%, respectively. Almost 65% (74/107) of persons had over 1% of their PBMCs infected of which 69.12% (47/68, six persons did not report gender) were female. Thirty-seven of these 47 women (78.72%) reported NHP exposures compared to 8/17 (47.06%) men with > 1% infected PBMCs. Five women and one man had > 10% infected PBMCs (range 11.77–73.06%), of which four women (66.7%) reported NHP exposure. We next examined the mean and median percentage of infected PBMCs in the 20 phylogenetic clusters consisting of two to 10 members. The overall mean and median percentages of infected PBMCs in these 20 clusters was 5.18% and 1.46% compared to 3.54% and 4.69% for singletons, respectively, but these differences were not statistically significant. The average and median percentages of infected PBMCs for potential transmission pairs, one three-member cluster, and the five- and eight-member clusters were 2.83% and 1.50%, 4.37% and 4.21%, 1.30% and 3.98%, 0.92% and 0.35%, 11.91% and 2.76%, respectively. Differences in mean and median percentages of infected cells in each cluster in comparison to those of single-tons were not significant. Two women in potential transmission clusters had the highest percentages (71.20% and 73.06%) of infected PBMCs compared to that for singletons (19.11%; a male). We did not find a difference in the average and median percentage of infected PBMCs between women (5.055% and 1.79%, n = 99) and men (2.78% and 1.47%, n = 46), respectively. Participants in the 0–5 yo age group had the highest mean and median percentages (19.74% and 2.95%, respectively) of infected PBMCs but this was likely skewed by the highest value (73.06%) in our study in a 15 yo female (MPX3146). Interestingly, the 8-member cluster with the highest mean percentage of infected cells (11.91%) contains LTR sequences with the 11-bp but again this is likely skewed by the high 73.06% results for the female member of this group (MPX3146).

## Discussion

Limited information exists on the current prevalence and characteristics of HTLV infection in DRC with most studies conducted in the 1990s [13, 32, 69–73]. To fill this knowledge gap, we conducted a cross-sectional, population-based survey among participants from two health zones in the rural Sankuru province of DRC to assess the prevalence, transmission risk factors, and biomarkers of HTLV infection in this population with high bushmeat exposure. We found an overall HTLV seroprevalence of 5.4% in our population and an HTLV-1-specific prevalence of 4.2%, further adding to our understanding of HTLV burden in the DRC. In comparison, previous prevalences reported in DRC ranged from 3.1–19.6%, excluding studies that focused on clusters of persons with HAM/TSP in which the prevalence was as high as 78.1% [13, 32, 69–73]. For example, a 1993 study of randomly sampled persons from the general population residing in Inongo, DRC, reported a crude HTLV-1 prevalence rate of 3.1% [72]. In Inongo, the authors report fish as the primary dietary protein, which may help explain the lower prevalence observed in this population compared to Sankuru, where NHP are a common food staple. The 19.6% seroprevalence was seen in adults from a leprosy hospital in Northwest DRC in the province of Mbandaka [70]. The most recent study reported in 2017 by Mossoun *et. al* in three villages in the Bandundu province of DRC, around 450 km west of Sankuru, reported a 1.3% prevalence rate for HTLV-1, though this sample only included 302 DRC individuals and only three persons were confirmed with infection by WB and PCR testing

[32]. Two HTLV studies in DRC's capital, Kinshasa, reported prevalence rates of 7.3% and 3.2% among sex workers, a population at higher risk for sexually transmitted infections, albeit less likely to have ubiquitous contact with NHP bushmeat [69, 74]. HTLV-associated pathologies such as ATLL and TSP/HAM have been difficult to estimate in DRC due to a lack of diagnostic facilities, trained medical personnel, and limited health system infrastructure [75]. Hence, research on HTLV-1-related diseases in DRC and across Africa is urgently needed and would help improve public health and disease prevention [3].

HTLV prevalence in sub-Saharan Africa varies across country, region, and ethnic group as a result of forest proximity, hunting activities, and other high-risk behaviors. A recent meta-analysis of HTLV-1 from published population-based studies in sub-Saharan Africa showed a higher seroprevalence in Central Africa (4.16%) compared to Western (2.66%) and Southern Africa (1.56%) [76]. This meta-analysis also found higher seroprevalence in women (3.27% vs 2.26%) and rural locations (3.34% vs 3.18%), congruent with our findings. Jeannel *et. al* also reported a higher seroprevelance in women (3.5% vs 2.6%) and showed the highest seroprevalence (6.5%) in the Bolia ethnic group in Inongo, DRC compared to 1.5% in the Sengele [77]. A more recent study in rural Gabon also reported a higher overall HTLV-1 prevalence rate (7.3%) with higher infection prevalence in women (9.0%) [77]. Taken together, the literature consistently shows an increased HTLV infection risk among women, especially as they age, matching our findings. Others have hypothesized that increased infection in women is likely from sexual transmission via condomless sex and higher viral loads in their male partners [1, 2, 41, 77]. Noteably, several of these studies were conducted in the 1990s and the lower specificity of HTLV assays at that time likely inflated these reported numbers [2].

Among HTLV-1 PCR-positive persons, three quarters were female compared to about 58% of the total study population. We previously reported more SFV infections in females in this population, including a woman with concurrent HTLV-1 infection, and identified an association of SFV seropositivity with butchering and skinning NHPs, which parallels the reported association of SFV infection with severe NHP bites in NHP hunters in Cameroon [33, 36, 40]. Some of these SFV-infected Cameroonians were also infected with HTLV-1 subtype B and F strains like those found in NHPs from Cameroon.

To better understand HTLV-1 transmission dynamics in our population, we conducted Bayesian phylogenetic cluster analyses. While most clusters were pairs, we also identified four clusters with three, five, and eight members each. As with our statistical analysis of HTLV seroreactivity, we found that most clusters consisted of females reporting NHP exposures and frequent forest activity. Importantly, we found that most transmission flowed across households, villages and health zones. Some of the villages in these clusters were 2 to 118 km apart along both major and local roads, suggesting possible HTLV transmission across long geographic distances and not just within and between proximal households. We only observed transmission within five households, including two male-female pairs indicating likely sexual transmission and two wife and child pairs which, based on age and household dynamics, could be indicative of vertical transmission, though the biological relatedness between wives and children was not assessed. Our findings are similar to those reported for HIV-1 in rural Africa that showed clusters of pairs within the same household that were connected to infections in other villages mostly via sexual contact with females [78]. The finding of mostly women in these clusters also suggests more opportunities for vertical transmissions and that we are likely missing important epidemiologic links. This is further supported by our finding of an equal number of singletons (persons not clustering) of which the majority were also women. Nonetheless, our finding of HTLV-1 disseminated across households, villages, and health zones indicate public health prevention programs at both the local and national levels are needed to interrupt transmission. As for HIV prevention, increased testing and educational

strategies with focused cluster detection and response efforts can help stem the spread of HTLV in these communities and may also help fill the prevention gaps identified here [79].

Phylogenetic analyses showed that our DRC HTLV-1 LTR sequences shared an evolutionary history with those from STLV-1. Most significantly, three STLV-1 LTR sequences from *A. nigroviridis* (STLV-1ang) from DRC were ancestral to most DRC HTLV-1 sequences from our study [58]. The three STLV-1ang sequences were obtained from captive *A. nigroviridis* from a zoo in Paris, France [58]. Reportedly, these three monkeys were the only seropositive animals at the zoo and were the offspring of an older dam from Central Africa, but the country was not provided. The habitat range of *A. nigroviridis* is in swamp forests in the Congo Basin and includes eastern Republic of Congo, western DRC where our study sites are located, and southern parts of the Central Africa Republic [80, 81]. *A. nigroviridis* are not yet an endangered species because swamp forests are not being logged or cleared but they are commonly found in the swamp trees and are easily hunted from boats in the Congo River and sold as bushmeat [80, 81]. Interestingly, *A. nigroviridis* has the highest prevalence of infection among seven STLV-1-infected NHPs tested in DRC and at 36.2% may be endemic in this species (S1 Table).

However, our phylogenetic results for this large clade of DRC HTLV-1 do not suggest multiple and recent STLV-1 introductions but rather a likely older introduction of STLV-1 that continued to spread in this population after becoming established in humans as a divergent HTLV-1 subtype B infection. This finding is analogous to the cosmopolitan HTLV-1a genotype for which a parental STLV-1 sequence has not yet been identified. Given that all HTLV likely originated from STLV, then HTLV-1a must have originated from a closely related STLV-1 after introduction into humans and then became endemic as for this HTLV-1 clade in DRC. Nonetheless, we did find STLV-1 sequences from apes and monkeys from DRC and Cameroon that phylogenetically clustered with our DRC sequences, but which were not strongly supported limiting our conclusions for these genetic relationships. Inclusion of additional STLV-1 sequences from DRC may help to resolve the origins of HTLV-1 in this population. Similar results have been reported in central Africa (Cameroon and Gabon) that showed the HTLV-1 in persons with severe NHP bite exposures did not always share a direct evolutionary history with STLV-1 including those from the same region and in persons with dual SFV infection [32, 33, 40]. However, our results are supported by the lack of an association of NHP exposures and HTLV WB positivity in our study suggesting community spread from person-to-person versus multiple primary zoonotic infections. While bonobos (*Pan paniscus*) in DRC are the only STLV-2-infected NHPs identified to date (S1 Table), and three persons in our study had positive HTLV-2 WB profiles, we did not observe any confirmed PTLV-2 infections in our study population [82]. Likewise, we did not find any HTLV-3 or HTLV-4 infections although STLV-3 is endemic in various monkeys in DRC but at lower prevalences than STLV-1 (S1 Table) [46].

We did, however, find evidence for dual SFV and HTLV-1 infection in three persons in our study [36]. Although all three persons (MPX8223, MPX21044, MPX40224) were infected with SFV most similar to NHPs endemic to DRC, including SFVcan from Angolan colobus and SFVasc from *C. ascanius* (red-tailed guenon) monkeys, all three HTLV-1s from these persons were within the large cluster of DRC sequences from our study that are potentially descendent from STLV-1ang. MPX8223 was also infected with the unique HTLV-1 strain identified in our study with an 11-bp deletion in the LTR region though this is likely unrelated to their SFV infection. Recently, it was shown that STLV-1 co-infection is associated with increased blood SFV pVLs and the authors showed that the STLV-1 *tax* protein can transactivate the SFV LTR to increase its replication [83]. While little is known about pVLs in dually infected humans, SFV pVLs in these three persons were within the range reported for SFV-infected humans and NHPs suggesting that their dual infections with HTLV-1 may not have affected their SFV pVL [36, 84, 85]. However, the HTLV-1 pVLs for MPX8223 and MPX21044 were both greater than

1%, and as described below, can indicate risk for HTLV-1-associated disease. Cross-sectional studies such as ours cannot discriminate which retrovirus infection, SFV or HTLV, occurred first in dually infected persons.

Previous studies have estimated that elevated HTLV-1 pVLs can be indicative of progression to disease or are found in persons with HTLV-1 disease, including ATLL and HAM/TSP, and can also increase the risk for person-to-person transmission. For example, persons with ATL and HAM/TSP have much higher pVLs compared to asymptomatic carriers and can also help predict disease progression in infected carriers [86]. Higher pVLs have also been associated with shorter survival times in ATLL patients [87]. A meta-analysis for HAM/TSP patients from the UK and US showed all had pVLs > 1% in PBMCs (> 100 proviral copies/$10^4$ PBMCs), though a definitive cutoff has not been established [88, 89]. Nearly 66% of HTLV-1-infected persons in our study had mean percentage of infected cells > 1% (range 1.1–73.2%) of which 69% were females. Interestingly, more women (n = 5) than men (n = 1) had pVLs > 10% (range 11.7–73.2%) though the mean percentage of infected PBMCs by gender and age were not significant despite more women testing PCR-positive in our study. While we did not record health status for our participants, our results indicate a large proportion may be susceptible to HTLV-1-associated diseases supported by previous studies that identified clusters of HAM/TSP in DRC [72, 73]. Indeed, the two women with extremely high percentages of HTLV-infected cells (71.2 and 73.2%) are more similar to the high pVLs seen in persons with ATLL than to those with HAM/TSP, which in one study had a mean of 50.3% median pVLs compared to 14.7%, respectively [86]. Our high pVL results are less than 100% suggesting that they are not due to multiple HTLV-1 integrations per PBMC cell which could complicate interpretation of the results and their potential association with transmission and/or disease [86]. We did not find a clear association of pVLs in potential transmission pairs or clusters identified on our study, except for the cluster containing LTR sequences with the 11-bp deletion that had the highest mean percentage of infected cells (11.9%) for clusters larger than two persons. This finding may reflect that the 11-bp deletion provides a viral replication advantage though the deletion occurs before the first transcription enhancer element in the LTR. It is possible the deletion changes the secondary structure of the LTR to increase replication though additional experiments are required to test this hypothesis. It should also be noted that this eight-person cluster contains the person with the highest percentage of infected cells which positively skews the mean percentage of infected cells in this group.

Overall, our epidemiologic findings were consistent with previous PTLV studies in DRC and reflect the challenges of studying a low incidence disease in participants with overlapping exposures to multiple animals. We found that rates of seroreactivity increased with each increasing age stratum suggesting continued exposure to PTLVs over the life course. Despite this, we were unable to detect a relationship between the nine individual NHP species included in our study questionnaire and HTLV seropositivity. Previous studies of PTLVs in Central Africa have found a near ubiquitous exposure to NHP, whereas our study found only 65% of participants reporting an NHP exposure in the past month [32]. This distinction might be explained by regional practices of Sankuru, our classification system, which focused on exposures from the previous month only, or other measurement errors arising from NHP misidentification by participants. We aimed to limit potential bias in NHP exposure reporting by providing pictorial representations of all species included in our study, rather than relying on name-based identification. Nonetheless, we also did not find any evidence of recent STLV-1 infection in our population despite high NHP exposure and previous identification of SFV infection in this same group. In the inferred phylogeny recent infection of persons from DRC in our study with STLV-1 would have been indicated by a direct link to an STLV-1, i.e. an HTLV-1/STLV-1 pair. Rather, our finding of only HTLV-1 infection in our study that is most

closely linked to other HTLV-1 from DRC likely suggests a more evolutionarily distant cross-species transmission that has since become established in this area.

Our study had several limitations. First, blood transfusions and injection drug use, two possible pathways for horizontal transmission, were excluded from the questionnaire due to their low prevalence in the community. Due to challenges of collecting robust biological samples among young children ages 0–5, this age group was underrepresented in the analytical sample. This may have made our observed prevalence of HTLV appear higher in the study population as the risk of viral infection was found to increase with age (lifetime exposure). Additionally, our capture of household family trees and completeness of biological data was based on convenience sampling of all household members present during sample collection and may likely explain the over-representation of women in our study and missing potential transmission linkages. Alternatively, some of the singletons could result from older infections in which the transmission link is lost or could reflect dead-end infections. We were also unable to obtain HTLV sequences and pVLs from all seroreactive persons which could have uncovered additional potential transmission linkages or phylogenetic relatedness to HTLV and STLVs and helped to further understand transmission and pathogenicity in our study population. The negative PCR results in these seroreactive persons could reflect low proviral loads, sequence divergence at the PCR primer binding sites, false-reactive serology results, and/or other factors. Indeed, commercial HTLV serology assays that only include HTLV-1 and HTLV-2 antigens have limited validation for detecting STLV-2, -3 and -4 which may limit their sensitivity for detecting these divergent viruses [47, 90]. Nonetheless, we PCR-tested all WB reactive samples with PTLV generic *tax* PCR primers and did not find HTLV-2, -3 or -4 in our study population. The addition of HTLV-3 and -4 specific antigens to existing serologic assays could help improve detection of these variants but must weigh the cost and public health benefits of doing so. In Sankuru, defining family relations presented challenges, particularly due to customs of polygamy and arrangements for which certain wives and their children may live in separate physical spaces overseen by a single head of household. In addition, due to the nature of household role and its inextricable link with both age and sex, a full explanatory model of the relationship between these three factors and HTLV serostatus could not be tested. This necessarily limits our understanding of how social structures, familial duties, and household clustering may impact person to person transmission of HTLV in the study population. Deep investigation into social networks within and between villages would be needed to trace transmission events with greater certainty, but our sequence analyses suggest evidence of some human-to-human transmission events consistent with the epidemiology of HTLV. Finally, although we identified high HTLV-1 pVLs in PBMC specimens from persons without a reported clinical diagnosis of disease, recent studies suggest that patient testing should also include pVL testing of cerebrospinal fluid or tissues from persons with suspected HAM-TSP or ATLL-lymphoma for diagnosis confirmation [86]. Furthermore, additional studies are required to determine pVL cutoffs to distinguish asymptomatic carriers from persons with HAM/TSP and ATLL and to standardize variation between assays as suggested [91].

Without the ability to determine a specific species or activity of concern, public health risk communication for zoonoses remains a challenge. Nonetheless, our results provide insight into the spread of HTLV-1 within and across distant villages, which requires clear HTLV-1 prevention communication and effective strategies at both the local and national levels as proposed to help eradicate HTLV-1 infection [3]. Further research is required to understand, why, if exposure is constant across the life course and occurring at a high rate, only some individuals may become seropositive for PTLVs. Future work must be done involving molecular typing of HTLV strains and STLV strains to help reveal zoonotic transmission links and further explore person-to-person transmission risks.

## Supporting information

**S1 Table. Animals in the Democratic Republic of Congo included in the study participant questionnaire.**
(DOCX)

**S1 BEAST. Phylogenetic analysis BEAST xml file that includes the study sequences and parameters used in the analysis.**
(XML)

## Acknowledgments

Use of trade names is for identification only and does not imply endorsement by the U.S. Centers for Disease Control and Prevention (CDC). The findings and conclusions in this report are those of the authors and do not necessarily represent the views of the CDC.

## Author Contributions

**Conceptualization:** Anne W. Rimoin, William M. Switzer.

**Data curation:** Megan Halbrook, Adva Gadoth, Nicole A. Hoff, William M. Switzer.

**Formal analysis:** Megan Halbrook, Adva Gadoth, Anupama Shankar, HaoQiang Zheng, Ellsworth M. Campbell, William M. Switzer.

**Funding acquisition:** Anne W. Rimoin, William M. Switzer.

**Investigation:** Anne W. Rimoin, William M. Switzer.

**Methodology:** Megan Halbrook, HaoQiang Zheng, Ellsworth M. Campbell, Anne W. Rimoin, William M. Switzer.

**Project administration:** Anne W. Rimoin, William M. Switzer.

**Resources:** Anne W. Rimoin, William M. Switzer.

**Software:** Anupama Shankar, Ellsworth M. Campbell, William M. Switzer.

**Supervision:** Jean-Jacques Muyembe, Emile Okitolonda Wemakoy, Anne W. Rimoin, William M. Switzer.

**Validation:** Megan Halbrook, Adva Gadoth, Anupama Shankar, Nicole A. Hoff, William M. Switzer.

**Visualization:** Megan Halbrook, Adva Gadoth, Anupama Shankar, William M. Switzer.

**Writing – original draft:** Megan Halbrook, Adva Gadoth, Anupama Shankar, Ellsworth M. Campbell, Nicole A. Hoff, Anne W. Rimoin, William M. Switzer.

**Writing – review & editing:** Megan Halbrook, Adva Gadoth, Anupama Shankar, HaoQiang Zheng, Ellsworth M. Campbell, Nicole A. Hoff, Anne W. Rimoin, William M. Switzer.

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
