## [Decision Letter · Decision Letter 0]

21 Sep 2020

Dear Dr. Switzer,

Thank you very much for submitting your manuscript "Human T-cell Lymphotropic Virus Type 1 Transmission Dynamics in Rural Villages in the Democratic Republic of the Congo with High Nonhuman Primate Exposure" for consideration at PLOS Neglected Tropical Diseases. As with all papers reviewed by the journal, your manuscript was reviewed by members of the editorial board and by several independent reviewers. In light of the reviews (below this email), we would like to invite the resubmission of a significantly-revised version that takes into account the reviewers' comments. 

We cannot make any decision about publication until we have seen the revised manuscript and your response to the reviewers' comments. Your revised manuscript is also likely to be sent to reviewers for further evaluation.

Sincerely,

Johan Van Weyenbergh

Associate Editor

Edgar Carvalho

Deputy Editor

Reviewer's Responses to Questions

**Key Review Criteria Required for Acceptance?**

**Methods**

-Are the objectives of the study clearly articulated with a clear testable hypothesis stated?

-Is the study design appropriate to address the stated objectives?

-Is the population clearly described and appropriate for the hypothesis being tested?

-Is the sample size sufficient to ensure adequate power to address the hypothesis being tested?

-Were correct statistical analysis used to support conclusions?

-Are there concerns about ethical or regulatory requirements being met?

Reviewer #1: see Summary and General Comments

Reviewer #2: The authors clearly stated the hypotheses and objectives to be achieved in this study. To this end, they have described the study population, whose size in numbers ensures that the conclusions drawn are sufficiently powerful. Meanwhile, The authors should clearly specify what was the original purpose of this population-based study and why they chose this particular region of the DRC and specifically these two villages? Are there any previously published results (with the exception of those concerning SFVs) from this population-based study?

Regulatory requirements appear to have been met and clarification regarding the procedure for anonymizing the samples was sought from the authors.

**Results**

-Does the analysis presented match the analysis plan?

-Are the results clearly and completely presented?

-Are the figures (Tables, Images) of sufficient quality for clarity?

Reviewer #1: see Summary and General Comments

Reviewer #2: It is important to describe the study population in more detail (mean and median age, overall range and for each gender). 

The presentation of the results should be modified to make it more understandable and comparable to studies that are conducted using a comparable methodology: detection of specific antibodies by ELISA and confirmation by WB and PCR. Therefore some tables will have to be modified to be more clearly presented.

**Conclusions**

-Are the conclusions supported by the data presented?

-Are the limitations of analysis clearly described?

-Do the authors discuss how these data can be helpful to advance our understanding of the topic under study?

-Is public health relevance addressed?

Reviewer #1: see Summary and General Comments

Reviewer #2: The conclusions drawn meet in part the objectives that have been defined. Indeed, some of the results could be analyzed and presented differently in a more rigorous and in-depth manner.

**Editorial and Data Presentation Modifications?**

Reviewer #1: see Summary and General Comments

Reviewer #2: We recommend that the authors make significant modifications (see Summary and General Comments section).

**Summary and General Comments**

Reviewer #1: The authors present a very interesting work on HTLV spread at a regional level in a rural area in the DRC. They combine several types of epidemiological data and methods to provide insight into the spread of HTLV-1 within and across distant villages. 

Most of my concerns relate to the phylogenetic analyses and their interpretation, which to my opinion merit some additional checks and better framing in the discussion section. 

#########

Abstract:

#########

- "Questionnaire data was analyzed with serological and molecular data."

=> Questionnaire data was analyzed in conjunction with serological and molecular data.

- "with most sequences sharing an ancestral history with STLV-1 "

=> not correct formulation

###############

Author summary:

###############

lines 99-100: "Divergent STLV-1, STLV-2, and STLV-3 have been reported in monkeys and apes in DRC, increasing the likelihood of exposure to these viruses [30-33]. " 

=> this sentence is not correct: the presence by itself does not increase the likelihood of exposure. Please adjust. 

###############

Methods: 

###############

- line 128: I am surprised to see such a high level of geographical clustering of the villages, which to me seems unexpected under a random sampling scheme. Can the authors comment on this please - was the selection truly random? 

- can the authors comment on the quality of the samples? I am not a wet-lab expert, but I can imagine that it is not easy at all to preserve samples in high quality conditions in remote areas, in that climate, and over such long time frames. Was a quality check on the sample integrity performed? Could sample degradation underlie for example that the reported estimates represent lower end estimates of the true HTLV prevalence? If so, this should be discussed.

- "Genetically related sequences were identified using BLAST searches (http://www.ncbi.nlm.nih.gov/BLAST/) and added to the analyses for comparison along with HTLV references"

=> the authors should be more clear as to how the dataset was compiled. How many BLAST hits were retained for each of the query sequences? Why was there an apparent need to separately add reference sequences and other HTLV sequence data from the DRC? 

- why did the authors choose for a constant-rate BD tree prior instead of a (non-parametric) coalescent model? Also, which priors were specified on the parameters of the BD-process? this should be specified, as well as the rationale for each. 

- fig 2b: The branch lengths seem quite close to a multiple of 1 - this suggests that there was very little diversity in the alignment, and that the temporal aspect may be unreliable. The authors should evaluate whether or not there is significant temporal signal in the data to calibrate the clock model. This can be done as in Trovao et al Virus Evolution, 2015, 1(1): vev016) by comparing the rate estimate with the expectation in the absence of temporal structure. Below the tweaks that need to be specified in the XML:

1) make sure that each taxon has an associated leafHeight parameter [this goes in the treeModel block]:

<leafHeight taxon="taxon1">

<parameter id="age(taxon1)"/>

</leafHeight>

 …

<leafHeight taxon="taxonN">

<parameter id="age(taxonN)"/>

</leafHeight>

2) add an operator that permutes the tip dates during the run, giving the evol rate expectation in the absence of temporal signal

<swapParameterOperator weight="10" forceGibbs="true">

<parameter idref="age(taxon1)"/>

 …

<parameter idref="age(taxonN)"/>

</swapParameterOperator>

3) to check whether all went well, log the tip-ages [and perhaps inlcude a tip age in the screen-logger element for a quick visual check]:

in the fileLog-element:

<parameter idref="age(taxon1)"/>

 …

<parameter idref="age(taxonN)"/>

and for the screenLog:

 <column label="age(taxon1)" sf="6" width="12">

<parameter idref="age(taxon1)"/>

</column>

The usual criterium is non-overlapping 95% credible intervals. The test for temporal signal should be reported, perhaps using a supplementary figure.

- The authors should justify their choice for the relaxed clock model: I believe that the HTLV evolutionary rate is linked to its transmission rate (Salemi, PNAS). If what the authors refer to as 'potential transmission clusters' indeed represent human-to-human transmission, I would think that one can expect 2 rate-distributions with different means operating on the phylogeny: a slower rate on the internal branches that represent evolution in the reservoir (on which I expect less frequent transmission), and a higher rate in clades that represent human-to-human transmission. In such situations the relaxed clock model will perform poorly (see a.o. Worobey et al Nature 2014), and a local clock model will be better suited. While the latter require prior knowledge on which branches to expect a rate difference, a random local clock model may offer a valuable 'naive' alternative (https://pubmed.ncbi.nlm.nih.gov/20807414/). Both can be set up using BEAUTi.

- the substitution model should be specified

- line 209: "significant ESSs" => reword to "high ESSs" (there is no criterium to bin ESS as significant or not to my knowledge)

- fig2a looks like an unrooted phylogeny, but it is not mentioned with which models/software this was inferred? 

- fig2b: this is not a nice presentation of the phylogeny. Better in this case is to use a rectangular layout. Also, the long tip labels hinder interpreting the tree - if these can be shortened to only contain the necessary information. The authors should use a time axis; why else estimate a time-calibrated phylogeny? The 2 major clades that are discussed should be clearly indicated. The purple for highlighting the clades with good support should be more bright to more clearly distinguis these clades from the rest. I miss a look at how long these cluster's MRCA goes back in time? Are they recent, old, a mixture of both? To when does the the 89-HTLV-clade MRCA date?

- about the potential transmission clusters: there clearly is intermixing of STLV with HTLV, suggesting rather frequent cross-species jumps. But is it no also true that the extant diversity of the STLV reservoir is severely underrepresented (undersampling of the STLV reservoir)? If so, what look like transmission clusters can very well be separate cross-species transmission events for which the source is simply not sampled. What arguments do the authors have against this scenario? They mention that 3 ang cluster as a sister clade to a 89-HTLV clade, but I don't see how this backs the idea that this 89-HTLV clade represents human-to-human transmission. 

A random local clock model can add evidence for either scenario: if there is a rate shift on the branch leading to the MRCA of the 89-HTLV clade such that this evolves more at a higher rate then the branches pertaining to reservoir evolution (of which I suppose less frequent transmission), this would be evidence against multiple cross-species transmissions.

- the NHPs in the questionnaire: many have not been shown to possibly harbour STLV-1 (supp table). I think that this substantially weakens the argument in favor of community spread based on "the lack of an association of NHP exposures and HTLV WB positivity" as NHP exposure via this measure does not very directly suggests exposure to STLV-1. 

Also, Can the lack of an association between NHP exposure and HTLV WB positivity be explained by higher sensitivty of the serological tests for HTLV-1 compared to HTLV2/3 (which are found in 4 of the NHP species)? This should be better discussed.

- what is the cluster definition? Only based on support? Why not also take some measure of time/genetic distance into account?

##########

Results

##########

- line 236: which levels? Pleeas specify in the Methods section, or detail here. 

- line 243: "Only a marginally significant association

244 between SFV and HTLV co-infection was observed (χ2 p-value= 0.0534).".

=> Is a Fisher exact test not better suited here? 

> m <- matrix(data = c(3,14,162,2872), nrow = 2, ncol = 2, byrow = T)

> m

 [,1] [,2]

[1,] 3 14

[2,] 162 2872

> fisher.test(x = m)

 Fisher's Exact Test for Count Data

data: m

p-value = 0.06041

alternative hypothesis: true odds ratio is not equal to 1

95 percent confidence interval:

 0.6925923 13.7944348

sample estimates:

odds ratio 

 3.795586 

###########

discussion:

###########

- "The overall HTLV seroprevalence in our population was 5.4% .... In accordance with our study’s 4.2% HTLV-1 prevalence"

=> contradiction?

- " Nonetheless, we also did not find any evidence of recent STLV-1 infection in our population despite high NHP exposure and previous identification of SFV infection in this same group."

=> I perhaps overlook it, but how was the recency of HTLV infection established?

#############

supplementary info:

#############

what does '0/1' under STLV's reported refer to? Please add this information.

Reviewer #2: In this manuscript, the authors studied the presence of HTLV infection in villages from a specific region of the Democratic Republic of the Congo (DRC).

It is important to perform such studies in DRC, a very highly populated country of central Africa where relatively few studies have been performed on such viruses in humans. This is an interesting paper that combines serological, viro-molecular and phylogenetic findings and provides some data on the transmission dynamics of HTLV-1 in remote villages in DRC with populations exposed to frequent contact with non-human primates.

This study was carried out by teams from the United States, one being headed by William M. Switzer, one of the leading scientists working on zoonoses in particular retroviruses such as HTLV and Simian Foamy Viruses.

We were therefore quite surprised by a certain lack of precision, particularly in the Methods and Results sections presented in this study. 

In this context, the quality of the manuscript should be greatly improved especially by answering the following remarks and questions.

Major comments: 

1) Population description. 

Page 7 lines 126 and following. 

-The authors should clearly specify what was the original purpose of this population-based study and why they chose this particular region of the DRC and specifically these two villages? Are there any previously published results (with the exception of those concerning SFVs) from this population-based study?

- In addition, they used only 3,051 of the 4,573 eligible individuals. Can the authors specify whether they think this fact may have introduced a bias in the study results? 

- What were the types of populations living in these areas (tribes, ethnic groups, ..)?

- Page 7, Line 136: Could the authors describe in more detail the anonymization procedure used for the samples they collected?

- How much blood did they sample from children, especially the younger ones, and what method did they use (EDTA tubes or DBS)?

2) Results. 

- It is important to describe the study population in more detail (mean and median age, overall range and for each gender). 

- It is not clear for the reviewer why the authors grouped the results after the WBs, considering the general HTLV item for sero-epidemiological analysis. Indeed, such HTLV includes, according to Table 1; HTLV-1, HTLV-2 and non-typeable HTLV. First, the authors should eliminate for their analysis those individuals diagnosed with HTLV-2 WB serology. Second, the results for non-typeable WB should be detailed: how many cases were HTLV-1/2 with reactivities to both MTA-1 and K55? It is very likely that in such a large series, some WB will exhibit such a pattern. Third, what does “like” mean after HTLV-1 and HTLV-2? According to the manufacturer of the WB kit, this is not mentioned. Finally, how many HGIP were observed among the 85 indeterminate pattern?.

- The authors should first focus their work on the HTLV-1 virus, which is by far the most prevalent and important virus in their population. 

- Another question is therefore to know what are the criteria to consider that a person HTLV-1 infected with HTLV-1 in their study: either the authors use a serological criterion only with HTLV-1 in WB and HTLV-1/2 including those for which the MTA- band 1 is stronger than the K55 band, or they also add a PCR criterion by including non-typeable profiles for which the HTLV-1 PCR (s) are positive. Authors should select and clearly define their positivity criteria.

Thus, the abstract, methods, results and Table 1 should be greatly modified based solely on analyses of HTLV-1 results.

This would provide more robust and comparable data with other studies carried out elsewhere in the world and particularly in Africa.

Subsequently, the authors may also describe data from the few individuals with HTLV-2 profile. It would be ideal to have specific HTLV-2 PCR as welle as others that can detect HTLV-3 and 4, as the authors know that people infected with the latter two retroviruses may have WB serologies, sometimes resembling HTLV-2. Can the authors comment on this, provide new data and modify their results?

Page 31, Table 1: In addition of the number of HTLV-1 positive samples, the authors should add the resulting prevalence of HTLV-1 in each category.

Page 32, Table 2: This table presents the odds ratio of being HTLV-1 in a household with at least one HTLV-1 infected member. Given that the prevalence of HTLV-1 increases with age and is higher among women, the results showed in this table are of limited interest and can lead to biased interpretation. Indeed, caution should be taken as the reference group here is the biological child. The odds ratio of being infected for the primary and secondary wife will obviously depend on which one is the mother of the child.

Page 32, Table 3: As demonstrated by the authors, the exposure to nonhuman primates and the type of exposure are significantly associated with gender. Additionally, there is a significant difference of HTLV-1 seroprevalence between sex. The analysis of NHP exposure as potential risk factors of HTLV-1 transmission should therefore be at least adjusted on sex, but can also be adjusted on both age and sex. Table 3 should therefore give sex-adjusted OR and the authors should add the number of HTLV-1 positive samples in each category.

Minor comments:

Introduction 

Page 4, Line 67: The authors can also quote here a recent paper published on the molecular epidemiology of HTLV-1 especially in Africa (Afonso et al., Retrovirology 2019).

Methods section

Study population sub-section

Questionnaire administration sub-section

Page 7, line 141: Regarding the data collected on exposure to non-human primates, how did the authors assess the bias related to unauthorized hunting activities that could result in the minimization of these activities?

Serology, PCR, and phylogenetic analysis sub-section

Page 8, Line 168: Considering the age of the samples collected in 2007, did the authors verify the quality of the DNAs by amplifying a housekeeping gene before carrying out targeted amplifications of the different HTLV genes/fragments?

Results 

NHP Exposer sub-section 

Page 12: To our knowledge, it is mainly the bite of an PNH that has proven to be the main ”zoonotic” risk factor for being infected with HTLV-1 in these village populations. Authors should isolate this factor in their analysis and not include it with "scratching" and "playing" activities.(see Table 3).

HTLV PCR and Phylogenetic Analyses sub-section 

Page 13: Concerning the PCR results, the authors must, for each serological profile in the WB (HTLV-1, HTLV-2, HTLV-1/2, HTLV not typeable, indeterminate including the HGIP profile,...), give the results of the different PCRs (specific for HTLV-1 at least, or also for HTLV-2, 3 and 4). This would make it possible to know the % of positive PCRs in each category. A table of this kind is available in the manuscript by Djuicy et al. PloS.TND, 2018, Table 1.

Page 14: To my knowledge, the first demonstration that Allenopithecus nigroviridis are infected by HTLV-1 genotype b, has been published by Meertens et al. in Virology in 2001. This paper describing the 3 strains included in the phylogenetic analysis (Angwis, Angmer, Angven) should be quoted and discussed in this article.

The different main clades of the phylogenetic tree (2B) should be presented more clearly with separations highlighting the different clades.

Pages 14-17: The section between lines 314 and 367 is long and often difficult to read. The authors might consider simplifying and shortening it.

Discussion

Page 18: In the introduction of the discussion, the authors could write a few lines on what is currently known about HTLV-1 related diseases in DRC (ATL, HAM / TSP, ..) and the situation of donors blood or pregnant women.

In general, authors should be cautious when referring to review articles especially for Africa and HTLV-1. Indeed, very often the methodologies used, such as the tests employed and the populations studied (pregnant women, blood donors, hospitalized patients, general population) are very different and not comparable. This applies in particular to the prevalence of HTLV on the African continent (see below).

Page 18, line 396: To our knowledge, it is not reasonable to write that the prevalence rate of HTLV on the African continent averages 10%. As the authors are well aware, it is almost impossible for HTLV-1 to give an average for a large region or even a country because of very different prevalence in different geographies. Regional microepidemiology distribution is a main feature of HTLV-1 distribution; this has been well demonstrated in parts of African and southern Japan. Moreover, in the general adult population, even in remote villages, the highest prevalence very rarely reaches 5 to 15%. 

As far as we know, this prevalence has only been observed in part of the rural population of Gabon. The data obtained in this study are quite comparable and constitute one of the highest prevalence of HTLV-1 observed in Africa. This finding could be clarified by the authors, at least in the abstract, results and discussion.Please comment on this topic and modify the sentence on line 396 accordingly.
---

## [Editor Report · Decision Letter 1]

26 Oct 2020

Dear Dr. Switzer,

We are pleased to inform you that your manuscript 'Human T-cell Lymphotropic Virus Type 1 Transmission Dynamics in Rural Villages in the Democratic Republic of the Congo with High Nonhuman Primate Exposure' has been provisionally accepted for publication in PLOS Neglected Tropical Diseases.

Best regards,

Johan Van Weyenbergh

Associate Editor

Edgar Carvalho

Deputy Editor

---

## [Editor Report · Acceptance letter]

8 Jan 2021

Dear Mr. Switzer,

We are delighted to inform you that your manuscript, "Human T-cell Lymphotropic Virus Type 1 Transmission Dynamics in Rural Villages in the Democratic Republic of the Congo with High Nonhuman Primate Exposure," has been formally accepted for publication in PLOS Neglected Tropical Diseases.

Best regards,

Shaden Kamhawi

co-Editor-in-Chief

Paul Brindley

co-Editor-in-Chief
